# Harnessing Macrophages through the Blockage of CD47: Implications for Acute Myeloid Leukemia

**DOI:** 10.3390/cancers13246258

**Published:** 2021-12-13

**Authors:** Luciana Melo Garcia, Frédéric Barabé

**Affiliations:** 1MD Anderson Cancer Center, Department of Stem Cell Transplantation and Cellular Therapy, University of Texas, Houston, TX 77030, USA; lmelo@mdanderson.org; 2Centre Hospitalier Universitaire de Québec—Université Laval, Québec, QC G1V 4G2, Canada

**Keywords:** CD47, acute myeloid leukemia, phagocytosis

## Abstract

**Simple Summary:**

The immune system is the first line of protection against infected and tumor cells. Macrophages are specialized immune cells that recognize these abnormal cells and eliminate them by a mechanism called phagocytosis. All normal cells express a protein called CD47 or “don’t eat me signal” to prevent their elimination through phagocytosis. Cancer cells, including leukemic cells, express higher levels of CD47 as a mechanism of protection against macrophage phagocytosis. CD47 blockade leads to an increase in phagocytosis of leukemic cells and better control of the disease. In this review, we explore CD47 function in normal conditions, its role in acute myeloid leukemia progression, and possible ways to block CD47 to enhance elimination of the leukemic cells improving the therapeutic options for patients with acute myeloid leukemia.

**Abstract:**

CD47 is a surface membrane protein expressed by all normal tissues. It is the so-called “don’t eat me signal” because it protects the cells against phagocytosis. The CD47 interacts with the signal regulatory protein alpha (SIRPα) on the surface of macrophages, leading to downstream inhibitory signaling that dampens phagocytic capacity. Since macrophages exert immune surveillance against cancers, cancer cells overexpress CD47 to defend themselves against phagocytosis. Acute myeloid leukemia (AML) is a cancer of hematopoietic stem/progenitor cells (HSPC), and similar to other types of cancers, leukemic blasts show enhanced levels of CD47. In patients with AML, CD47 has been associated with a higher disease burden and poor overall survival. Blockage of CD47-SIRPα signaling leads to improved phagocytosis of AML cells and better overall survival in xenograft models. However, the introduction of a pro-phagocytic signal is needed to induce greater phagocytic capacity. These pro-phagocytic signals can be either Fc receptor stimulants (such as monoclonal antibodies) or natural pro-phagocytic molecules (such as calreticulin). Based on these pre-clinical findings, various clinical trials investigating the blockade of CD47-SIRPα interaction have been designed as monotherapy and in combination with other anti-leukemic agents. In this review, we will discuss CD47 biology, highlight its implications for AML pathophysiology, and explore the potential clinical translation of disrupting CD47-SIRPα to treat patients with AML.

## 1. Introduction

CD47 is a transmembrane protein ubiquitously distributed in tissues, including erythrocytes, platelets, and hematopoietic stem cells. It is expressed by both normal and malignant cells being initially recognized as an ovarian tumor protein [1]. The first reports on this 50 kD multidomain glycoprotein described its association with integrins, and CD47 was originally named integrin-associated protein (IAP) [2]. In 1994, Mawby et al. finally integrated these findings by showing that CD47 shared the same structure as the previously described IAP and the ovarian tumor protein [3]. The vast distribution of CD47 suggested its possible implication not only in diverse physiologic functions but also in pathologic situations, such as in acute myeloid leukemia (AML).

Following its structural characterization, CD47 role as a “marker of self” or “don’t eat me signal” was shown to be dependent on its interaction with signal regulatory protein alpha (SIRPα), an inhibitory receptor in the surface of macrophages, dendritic cells, and neutrophils [4]. CD47-SIRPα interaction on the surface of macrophages results in the intracellular activation of immunoreceptor tyrosine-based inhibitory motifs (ITIMs) and phosphorylation of src homology 2 domain-containing phosphatases (SHP-1 and SHP-2), leading to diminished interaction between actin and myosin- 2 [5]. This cascade of events culminates in the inhibition of phagocytosis. Furthermore, the deletion of CD47 in red blood cells leads to their elimination in the spleen by macrophages through the activation of phagocytosis [6]. Thus, CD47 is a molecule capable of protecting cells against phagocytosis in steady-state conditions.

CD47 is not found only in normal cells. Malignant cells, such as leukemic cells, overexpress CD47 to avoid phagocytosis as an immune escape mechanism. This feature has been explored to develop new therapies targeting this so-called “phagocytosis checkpoint” [7]. Hence, molecules that inhibit CD47- SIRPα interaction might improve anti-leukemic control, especially in synergism with other therapies that trigger phagocytosis [8].

In this review, we will explore the implications of the CD47-SIRPα axis in AML pathophysiology, highlighting the novel therapeutic opportunities that exploit the blockage of this “don’t eat me” checkpoint.

## 2. CD47 Structure and Its Ligands

CD47 is a surface receptor expressed in all normal cells and tissues, including erythrocytes, lymphocytes, granulocytes, monocytes, platelets, and hematopoietic stem/progenitor cells (HSPCs) [2]. It is constituted by an extracellular immunoglobulin (Ig)-like portion, a pentaspan transmembrane domain, and a short intracytoplasmic tail [3,9] (Figure 1). Unlike other members from the immunoglobulin superfamily (which dispose of a single transmembrane domain), CD47 has an unusual conformation because its single Ig-like moiety is linked to a pentaspan transmembrane portion. This transmembrane domain seems to be implicated in several signaling pathways initiated by CD47 activation. Besides, its highly glycosylated Ig-like portion is essential to the interaction with its ligands and its association with integrins in a cis manner in the plasma membrane. Finally, its C-terminus intracytoplasmic tail lacks enzymatic activity and can be post-translationally modified by alternative splicing giving rise to four isoforms of the protein [10].

Initially described as an “integrin-associated protein” (IAP), CD47 physically and functionally associates with integrins in the plasma membrane [2], where they can form stable complexes [11]. Through an inside-out signaling, activation of CD47 leads to integrin affinity modulation and enhances integrin capacity to bind to its own ligands such as collagen, fibrinogen and vitronectin. This is the case for the integrins αvβ3 [12], αIIbβ3 [13] and α2β1 [11]. For instance, CD47 (in a cis manner) is essential to the platelet spreading that happens upon activation of integrin αIIbβ3 [13]. The relevance of CD47-integrin association has also been shown in an osteoarthritis murine model and patients with osteoarthritis. In both mice and humans with osteoarthritis, there is overexpression of CD47 and αvβ3 integrin, induced by the products of damaged chondrocytes. Cooperative signaling derived from the CD47-αvβ3 integrin complexes leads to activation of synoviocytes and monocyte-derived macrophages, resulting in enhanced inflammatory responses [14]. These data show that CD47-integrin association has implications in physiological and pathological contexts.

Another CD47 ligand is thrombospondin-1 (TSP-1). TSP-1 is a matrix glycoprotein secreted by several cell types such as monocytes, macrophages, dendritic cells, adipocytes and fibroblasts [15]. It plays an essential role in the modulation of inflammation as a potent inducer of transforming growth factor (TGFβ-1) [16], an anti-inflammatory molecule. Besides, it influences immune responses through its active binding domains that allow its interaction with diverse receptors, including lipoprotein receptor-related protein (LRP1), CD36 and CD47. TSP-1 binds to CD47 through its C-terminal domain and their interaction seems to modulate immune responses in the context of inflammation. On leukocytes, CD47-TSP-1 intercommunication results in the down-modulation of immune subsets, which might be essential in the context of inflammation [17]. Furthermore, TSP-1 is the molecule responsible for the activation of CD47-integrin complexes acting through CD47 to facilitate integrin activation [12].

The most studied CD47 ligand is SIRPα (also named CD172a or SHPS-1) [4], an inhibitory receptor expressed on the surface of myeloid cells such as monocytes, macrophages, granulocytes, and dendritic cells [18,19]. Other members of the SIRP family, such as SIRPβ and SIRPγ, also bind to CD47, but the importance of these interactions remains unclear [18,20]. CD47 interaction with SIRPα is considered weak, species–specific, and inversely proportional to the level of glycosylation of the SIRPα extracellular domain [21]. Its extracellular domain comprises three Ig-like portions, which are linked to a single transmembrane region and an intracytoplasmic moiety containing ITIMs (Figure 1). Initially named SHPS-1 due to its binding capacity to both SHP-1 and -2 tyrosine phosphatases, SIRPα signals through its two ITIMs to recruit and activate cytosolic SHP-1 and -1 leading to downstream dephosphorylation of signaling pathways [5,22]. It results in an inhibitory stimulus to the cells expressing SIRPα and dampens phagocytosis in myeloid cells [5]. Upon ligation of CD47, SIRPα localizes into the immune synapse leading to inactivation of integrins on macrophage surface and decreased phagocytic capacity [23] (Figure 2).

CD47-SIRPα interaction happens through their respective Ig-like domains [19]. This interaction is unusual because Ig-like domain bindings typically happen in a two-to-two instead of three-to-one way, the latter being for one single CD47 and three SIRPα Ig-like portions [24]. The arrangement of four Ig-like domains still allows for the necessary distance to form an efficient immune synapse [25]. Of interest, a disulfide bond between the Ig-like and the transmembrane portions seems to be essential to CD47-SIRPα signaling [26]. CD47 binding to SIRPα has been well studied because SIRPα is a phagocytosis checkpoint, and its modulation has significant clinical applications, such as for the treatment of patients with AML and other hematologic malignancies.

## 3. Role of CD47 as a Phagocytosis Checkpoint in AML

### 3.1. CD47 Overexpression in AML

CD47 induces a non-phagocytic signal upon ligation to its receptor SIRPα on the surface of macrophages. It is overexpressed by cancer cells as an immune escape mechanism, and leukemic blasts also depict this protection against phagocytosis.

Murine leukemic blasts express high levels of CD47 [27,28]. Interestingly, leukemic blasts from either primary or secondary transplanted mice show a 3–20-fold increase in their levels of CD47. This CD47 enrichment in peripheral blood (PB) blasts is found both in the hematopoietic stem cell (HSC) and progenitor stages, except for the megakaryocytic/erythroid restricted progenitors [29].

Similar to murine leukemic stem cells (LSCs), human LSCs and progenitors express higher levels of CD47 when compared to normal controls, including human bone marrow and cord-blood CD34+ HSCs together with mobilized PB HSCs. Besides, CD47 expression positively correlates with the engraftment capacity of human leukemic cells in mouse models, possibly due to the attenuated phagocytic potential of these CD47 high-expressing LSCs [29]. Not only human LSCs overexpress CD47, but AML genomic subgroups have diverse CD47 expression with FLT3-ITD mutated AMLs displaying greater CD47 levels while favorable risk AMLs, such as t(8;21), depicting lower levels of the “don’t eat me signal” molecule [30].

In an immunohistochemistry-based study, bone marrow (BM) samples of 171 newly diagnosed patients with primary AML were stained for the expression of CD47. Using a grading system to determine the level of CD47 expression, nine (5.2%) of these patients did not express, 130 (76%) had a low or intermediate level, while 32 (18.7%) expressed a high level of CD47. On the other hand, only 4% of patients with secondary AML overexpressedCD47. Unlike Majeti et al., no correlation between CD47 expression and FLT3-ITD mutation or t(8;21) was found in this cohort. Instead, CD47 expression positively correlated with NPM1 mutation AML [31].

Methodological differences might have caused the discrepancies found by Majeti et al. and Galli et al. The first group focused on the flow cytometric analysis of PB blasts and LSCs, and the later IHC study was performed in BM non-mobilized leukemic cells, a distinct micro-environment. These malignant cells within the BM display close contact with the BM stromal tissue and are exposed to a particular cytokine profile. They may hence phenotypically differ from PB blasts. Besides, HSC mobilization from the bone marrow is associated with greater CD47 expression, most likely to minimize the phagocytosis of these cells in the periphery. Thus, their findings might not necessarily be controversial but caused by distinct methodological approaches.

Finally, the expression of CD47 has been studied in conditions that might evolve into AML, such as myelodysplastic syndrome (MDS), myeloproliferative disorders, and chronic myeloid leukemia (CML). Patients that progressed from CML to AML displayed enhanced levels of CD47, but not patients with stable chronic phase CML. In the same study, individuals with myeloproliferative disorders did not show an increase in CD47 expression [29]. In individuals with high-risk MDS, CD47 expression is elevated compared to low-risk or normal controls [32]. These findings suggest that CD47 overexpression might be a late event in the malignant evolution of AML.

### 3.2. Impact of CD47 Expression in AML Prognosis

Due to its overexpression in leukemic blasts and progenitors, CD47 expression has been studied as a prognostic marker in AML. In both an exploratory and a validation cohort of AML patients with various cytogenetic and molecular abnormalities, high CD47 surface expression was a marker of poor prognosis, leading to lower event-free and overall survival [30]. In an IHC-based study, the expression of CD47 positively correlated with disease burden, with higher CD47 levels being associated with a more significant disease burden. However, patients with higher CD47 expression did not depict a worse prognosis [31]. Furthermore, CD47 might confer survival advantages to leukemic cells, which are neither related to the immune escape mechanism nor the protection against phagocytosis. First, CD47 expression in leukemic blasts has been related to a chemoresistant phenotype [33]. Second, the transitory disruption of CD47 expression in leukemic blasts leads to a decrease in anti-apoptotic markers with consequent apoptosis [34]. Thus, CD47 might provide leukemic blasts with biological advantages that improve their endurance and enhanced resistance to therapy.

The impact of CD47 overexpression on AML biology and disease outcome brought up the rationale for developing the treatments that disrupt the CD47-SIRPα interaction. Since its overexpression might be related to an immune escape mechanism against phagocytosis and might confer survival advantages, anti-CD47 therapies have been studied and have shown promising results in individuals with AML.

### 3.3. How Anti-CD47 Therapies Improve Phagocytosis

Macrophages are professional phagocytic cells, and they rely on this feature to perform cancer immune surveillance. These innate immune cells engulf cancer cells in order not only to eliminate them but also to present their antigens and activate the anti-tumor adaptive responses [7].

For phagocytosis to happen, macrophages must recognize cancer cells as foreigners, which depends on a balance between pro- and anti-phagocytic signals. Pro-phagocytic signals result from naturally occurring molecules, such as calreticulin, or Fc receptor-derived signals that arise from immunoglobulins, including administered monoclonal antibodies such as the anti-CD20 rituximab [7]. Calreticulin is a widely expressed multifunctional protein, initially thought to be present only within the endoplasmic reticulum. However, this protein can be transferred from the endoplasmic reticulum to the surface membrane, where it acts as a pro-phagocytic signal in apoptotic and cancer cells [35,36]. Calreticulin binds to the low-density receptor-related protein-1 (LRP1) on the surface of macrophages [35] and is the primary pro-phagocytic signal that counteracts CD47 in cancer cells, being essential for phagocytosis during the disturbance of CD47-SIRPα interaction [36].

Anti-phagocytic signals are often overexpressed in cancer cells to counterbalance macrophage surveillance, and the most studied “don’t eat me” signal is the checkpoint molecule CD47. Various mechanisms explain how CD47-SIRPα intercommunication dampens phagocytosis (Figure 2). First, the interaction between CD47 on cancer cells and SIRPα on the surface of macrophages leads to diminished cytoskeleton activity, which is reflected by decreased non-muscle myosin and actin in the phagocytic synapse [37]. Second, the inhibitory signals generated by SIRPα and its downstream phosphatases, SHP-1 and −2, dampens tyrosine phosphorylation from Fc receptor activation, consequently blocking phagocytosis. Finally, the presence of SIRPα in the phagocytic synapse, caused by its interaction with CD47, seems to be essential because it inactivates integrins leading to a decrease in macrophage adhesion and spreading through the phagocytic surface [23]. Combining these negative signals culminates in a decrease in macrophage phagocytic capacity, which is why the overexpression of CD47 confers survival advantages to cancer cells.

Anti-CD47 therapies block CD47-SIRPα interaction and restore phagocytosis, consequently increasing macrophage-related anti-tumor activity. In the context of AML, the addition of anti-CD47 monoclonal antibodies leads to enhanced phagocytosis without induction of direct apoptosis in leukemic blasts [30]. Besides, CD47 blockade causes decreased engraftment of leukemic stem cells in immunodeficient mice, suggesting that CD47 might be essential for successful engraftment of AML. Administration of anti-CD47 monoclonal antibodies in mice with established AML leads to depletion of leukemic cells in PB and BM together with improved survival [30]. These CD47 blocking antibodies also improve adaptive responses by enhancing antigen presenting capacity within the tumor microenvironment, promoting CD8+ T cell anti-tumor cytotoxicity [38,39]. Based on these pre-clinical data, new anti-CD47 therapeutic approaches have been brought to the clinic to treat individuals with AML. Table 1 shows a summary of the clinical studies investigating the CD47 blocking agents.

## 4. Trials with CD47 Blocking Agents for Treatment of AML

### 4.1. Hu5F9-G4: A Humanized Anti-CD47 Monoclonal Antibody

The Hu5F9-G4 (5F9; magrolimab) is a humanized monoclonal antibody with enhanced affinity to CD47 and was developed for the treatment of malignancies, including AML. It has been generated as an IgG4 scaffold to minimize toxicity because IgG4 immunoglobulins are less likely to recruit Fc-derived antibody-dependent cellular phagocytosis (ADCP) or complement-dependent cytotoxicity (CDC) of normal CD47-expressing cells (Figure 3). Besides, its heavy chain constant region displays a Ser228Pro substitution lessening the Fab arm exchange, characteristic of IgG4 antibodies. In preclinical studies, it effectively increases phagocytosis of lymphoma cells, solid tumors, and primary AML samples [30,40].

Against lymphoma cells, 5F9 displays synergistic activity in combination with rituximab, an anti-CD20 monoclonal antibody that induces a robust pro-phagocytic signal and elicits phagocytosis [41]. A phase Ib study explored the administration of these two monoclonal antibodies to treat patients with relapsed or refractory CD20-positive lymphomas. Fifteen individuals with diffuse large B cell lymphoma and seven with follicular lymphoma received a priming dose (1 mg/kg) followed by a maintenance regimen of 5F9 in a dose-escalating manner. The 5F9 was administered until disease progression, clinical futility, or limiting toxicities occurred. All patients received rituximab at a dose of 375 mg/m^2^ of body surface area for weekly doses during cycle 1 in weeks 2, 3, and 4 and then monthly for six months. A transient expected on-target anemia occurred in 41% of patients due to selective elimination of senescent red blood cells in the circulation expressing higher pro-phagocytic signals. This adverse effect was mitigated by administering a priming dose of 5F9 followed by a maintenance regimen during which complete recovery of hemoglobin levels was observed. Only three patients required red blood cell transfusions. Half of the patients responded to treatment, and 36% experienced complete responses. Better responses were seen among patients with follicular lymphoma [8].

In a phase I dose-escalation study, 62 patients with diverse solid cancers and aggressive lymphomas received 5F9. A priming dose was tested during the phase Ia, followed by a dose-escalation of a maintenance regimen (1b) and a loading dose on day 11 of the first cycle (phase Ic). The drug was considered to be safe despite the wide distribution of CD47 in normal tissues. Cytopenias occurred in more than 30% of patients, and anemia was the most often hematologic adverse event found. This on-target anemia was transient and characterized by a decrease in hemoglobin within the first two weeks of the priming dose. Marked reticulocytosis was observed after 15 days, and recovery of hemoglobin levels was seen after 5 to 6 weeks of the first dose, even in patients on maintenance therapy. Red blood cell transfusions were necessary for only four patients. No patients developed grade 3 or 4 thrombocytopenia, and hemagglutination occurred in 41% of patients without significant clinical repercussions. Infusion reactions, fatigue, headaches, fever, and hyperbilirubinemia were also frequent adverse events and were seen in more than one-third of the study participants [42].

In the context of myeloid malignancies, 5F9 is being tested both as monotherapy and in combination with azacytidine. Azacytidine, a hypomethylating agent, enhances the expression of calreticulin on the surface of AML cells in vitro and elicits phagocytosis in the presence of CD47 blockade. In immunodeficient mice, the combination of azacytidine and 5F9 leads to increased clearance of AML and improved survival of these animals [43]. Based on these preclinical data, the administration of 5F9 either in monotherapy or in combination with azacytidine is currently ongoing for patients with AML or MDS (NCT03248479). This safety phase Ib clinical trial is now recruiting patients with relapsed or refractory AML or MDS to receive 5F9 alone, and patients with untreated AML who are ineligible to induction chemotherapy or patients with high-risk MDS to receive 5F9 plus azacytidine. Both 5F9 alone and combined with azacytidine were well tolerated, and a maximum tolerated dose was not reached [44]. Anemia, fatigue, neutropenia, thrombocytopenia, and infusion reactions were the most common adverse events, and were found in 38%, 21%, 19%, 18%, and 16% of patients, respectively. Only one patient discontinued treatment due to side effects. Among transfusion-dependent patients, 58% and 64% of patients having MDS and AML became transfusion independent, respectively. Ninety-one percent of patients with MDS had an objective response, including 42% who achieved complete response and 24% who had complete bone marrow response. Among 64% of patients with AML, an objective response was observed, 40% being a complete remission (CR) and 16% complete remission with incomplete count recovery (CRi). The median duration of response was not reached for any of the groups [45]. Based on these encouraging results, a phase III double-blinded clinical trial, the ENHANCE study (NCT04313881), was designed and has started recruiting. This trial will investigate the efficacy of 5F9 in combination with azacytidine versus azacytidine alone for untreated patients with high-risk MDS.

Furthermore, two other studies are currently investigating the combination of the humanized monoclonal antibody 5F9 with diverse anti-leukemic agents. The first one is a phase Ib/II study recruiting patients with untreated AML who are not suitable for intensive induction chemotherapy (NCT04435691). They will receive 5F9 together with azacytidine and venetoclax, and treatment will be continued until disease progression or limiting toxicity. The primary outcome will be the maximum tolerated dose of this regimen. The other study is a phase I/II multi-arm clinical trial that will evaluate the safety and tolerability of 5F9 in combination with diverse anti-leukemic agents (NCT04778410). In cohort 1, patients with untreated AML ineligible to induction chemotherapy will receive venetoclax and azacytidine. Patients with relapsed or refractory AML will be part of cohort 2 and will receive mitoxantrone, etoposide, and cytarabine, while patients in CR or CRi with minimal residual disease after induction (cohort 3) will receive CC-486, an oral version of azacytidine. Hence, these studies highlight the potential of CD47-SIRPα interaction blockade to treat patients with myeloid malignancies such as AML and MDS, a population that lacks effective therapeutic options.

### 4.2. TTI-621: A SIRPα-IgG1 Fusion Protein

The TTI-621 is a soluble fusion protein comprising the CD47-binding N-terminal domain of SIRPα and the human IgG1 Fc region of IgG1 (Figure 3). The first portion binds to CD47 on the surface of tumor cells blocking its interaction with SIRPα, which hampers the “don’t eat me signal” within macrophages. The latter region interacts with Fc receptors on the surface of these phagocytic cells and elicits potent pro-phagocytic responses. Hence, this molecule circumvents the inhibitory signal generated by CD47 overexpression and enhances phagocytosis through Fc receptor stimulation [46].

This decoy protein interacts with CD47 on the surface of tumor cells, leading to increased phagocytosis of AML and MDS cells in vitro through engagement of Fc receptors. Interestingly, TTI-621 does not cause phagocytosis of normal cells despite its binding to leukocytes of healthy donors and shows minimal binding to red blood cells, which indicates a specific anti-tumor activity. Furthermore, TTI-621 infusion leads to significant anti-tumor activity against AML in xenograft models [46]. These findings support the potential of TTI-621 as an effective anti-leukemic agent.

These pre-clinical results guided the development of a phase I clinical trial to investigate the safety of the fusion protein TTI-621 in patients with relapsed or refractory hematologic malignancies (NCT02663518). The dose-escalation portion of this study (3 + 3 design) included patients with refractory lymphomas and aimed to evaluate the safety, maximum tolerated dose, pharmacokinetics, and pharmacodynamics of TTI-621. Its dose-limiting toxicities were grade four thrombocytopenia needing platelet transfusion and transitory alanine/aspartate aminotransferase elevations. The dose-expansion group included 146 patients with diverse hematologic and solid cancers, including 20 (14%) with AML and 6 (4%) with MDS. Treatment-related adverse events occurred in 131 (80%) participants, the most common being infusion-related reactions (43%), thrombocytopenia (26%), chills (18%), fatigue (15%) and anemia (13%). Infusion reactions were not dose-limiting and were mainly seen during the first infusion. The on-target thrombocytopenia was transitory, not dose-related, and occurred within the first four hours after administering the drug with rapid recovery over the following week. Bleedings of any grade occurred in 10 (6%) of patients, and grade > or =3 epistaxis occurred in 2 (1%) patients, one who had AML and the other MDS [47].

Regarding TTI-621 efficacy for patients with myeloid malignancies, 18 patients with AML and high burden disease showed progressive disease. Two patients with AML entered the trial bearing complete hematologic responses. One developed complete molecular remission and the other maintained stable disease throughout the study. The patients with MDS had either stable or progressive disease after administration of TTI-62147 [47]. In this study, TTI-621 had low activity against myeloid malignancies, a surprising occurrence because CD47 overexpression is likely a significant immune escape mechanism in AML. This lack of efficacy might have been due to the low doses of TTI-621 in the AML population to avoid drug-related thrombocytopenia.

### 4.3. CC-90002: An IgG4 Monoclonal Antibody

The CC-90002 is a high affinity anti-CD47 antibody built on an IgG4 scaffold to minimize toxicity related to on-target action on normal cells (Figure 3). It recognizes a CD47-specific epitope without causing hemagglutination. Besides, its IgG4 portion has two mutations (S228P and L235E) that dampen CC-90002 affinity to Fc receptors, leading to minimal ADCC and CDC cytotoxicities. These IgG4 mutations were introduced to decrease the risk of elimination of normal cells that also express CD47. In vitro, CC-90002 increased phagocytosis of AML cell lines and primary AML samples. Furthermore, CC-90002 showed activity against leukemic cells in xenograft models and led to improved survival of immune-compromised mice [48].

These findings led to the outline of a phase I clinical trial to investigate the administration of CC-90002 in patients with AML and MDS (NCT02641002). The dose-escalation phase (3 + 3 design) recruited patients with relapsed or refractory AML and patients with high-risk MDS. The results were not sufficiently encouraging, and the study was terminated without progressing to the expansion portion.

## 5. Conclusions

The CD47 is a widely expressed protein that protects cells against phagocytosis. It interacts with SIRPα on the surface of macrophages leading to an inhibitory signal that dampens phagocytic activity. Overexpression of CD47 is observed in AML and is a mechanism of immune evasion. Hence, the blockade of CD47 can be a strategy to harness the innate immune system against leukemic cells improving AML treatment. In preclinical studies, CD47-SIRPα inhibition resulted in improved phagocytic activity and better control of leukemia. However, challenges that impact the results down the clinical pipeline remain for the appropriate study of CD47-SIRPα blockade. First, most CD47-SIRPα studies are performed in immune-deficient mice, which hardly mimic macrophage activity or the tumor microenvironment. So, current xenograft models are far from ideal and might not correspond to the drug activity in humans. Second, to promote phagocytosis, a concomitant pro-phagocytic stimulus is needed because solely the blockade of the “don’t eat me signal” is not enough. Thus, CD47 blockade should be combined with other monoclonal antibodies that strongly activate phagocytosis through the Fc receptors or with molecules that increase the expression of calreticulin on the surface of tumors stimulating macrophages through the LRP1 receptor. This last strategy seems to be successful in patients with AML since the combination of CD47 and azacytidine has been shown effective without the addition of unmanageable toxicities. Thus, blockade of the phagocytosis checkpoint CD47 is a promising approach, especially in combination therapy for patients AML and MDS, a population still requiring effective therapeutic options.

## Figures and Tables

**Figure 1 cancers-13-06258-f001:**
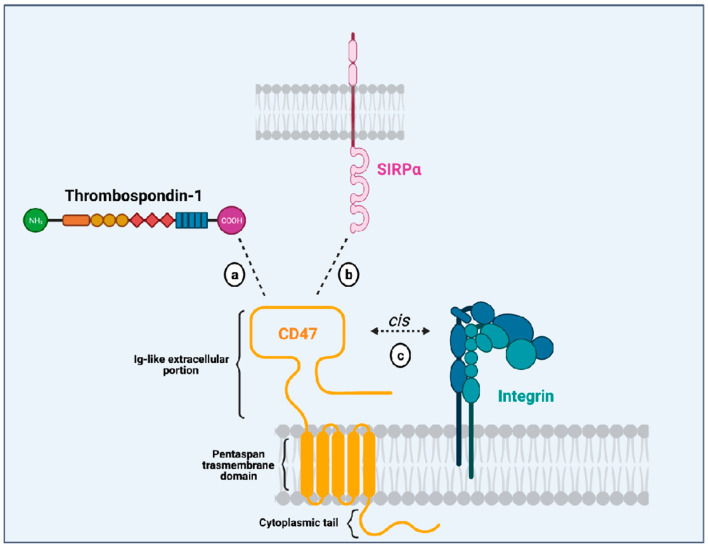
Illustration of the CD47 structure (in yellow) showing its single immunoglobulin (Ig)-like domain that interacts with thrombospondin-1 (TSP-1; (**a**), SIRPα (**b**), and integrins (**c**). It also depicts CD47 multispan transmembrane domain and its intracytoplasmic tail. Created with Biorender.Com.

**Figure 2 cancers-13-06258-f002:**
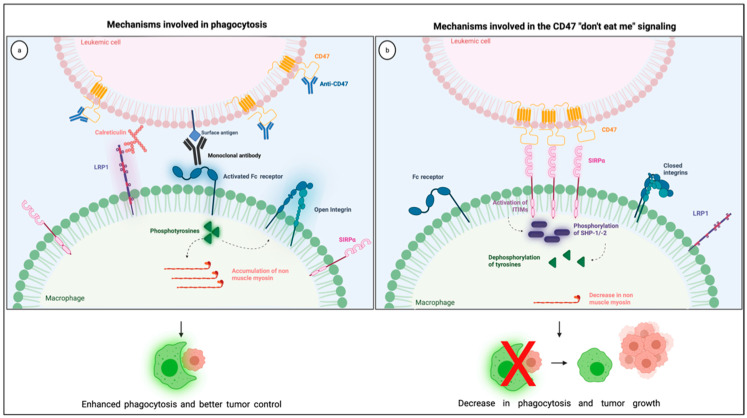
Illustration showing the mechanisms involved in the CD47 blockade and pro-phagocytic signals leading to phagocytosis (**a**) together with the mechanisms involved in the CD47 “don’t eat me signaling” through its interaction with SIRPα in the phagocytic synapse (**b**). Created with Biorender.com.

**Figure 3 cancers-13-06258-f003:**
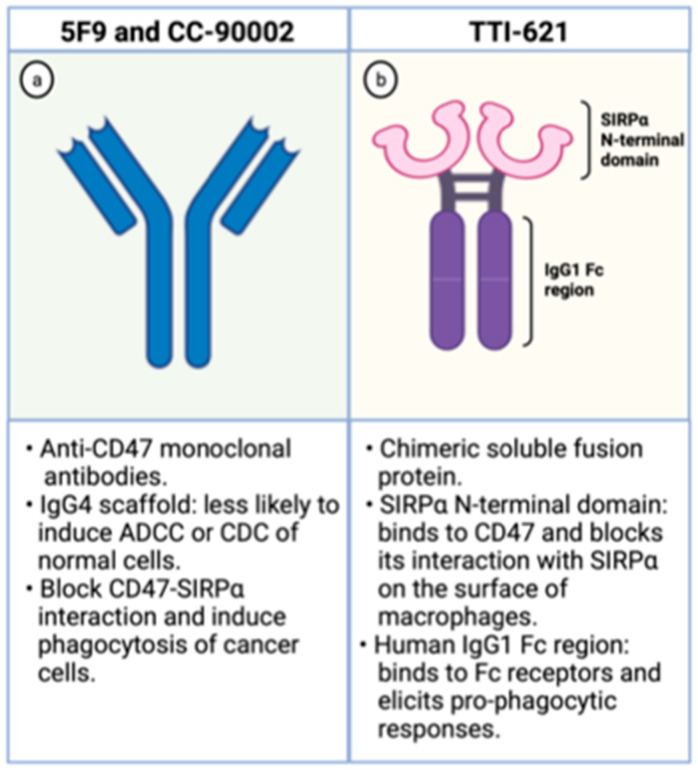
Schematic representation of CD47 blocking agents. On the **left** (**a**), schematic representation of anti-CD47 monoclonal antibodies. On the **right** (**b**), schematic representation of TTI-621, a chimeric fusion protein composed by the SIRPα binding domain and the human IgG1 Fc region. Created with Biorender.com.

**Table 1 cancers-13-06258-t001:** Overview of the clinical studies using CD47 blocking agents for the treatment of hematologic malignancies.

First Author (Journal, Year)/NCT Number	Drug	Combination Therapy	Phase	Patient Population	Status	Results	Most Common Adverse Events	Hematologic Toxicities
Advani (NEJM, 2018) NCT02953509	5F9	Rituximab	Ib	DLBCL and follicular lymphoma (Relapsed disease)	Completed	Objective response ^#^ in 50%; CR in 36%.	Fatigue (55–60%), headache (41%), infusion-related reactions (36%)	Transient anemia (41%), thrombocytopenia (15%), neutropenia (20%)
Sikic (JCO, 2019) NCT02216409	5F9	No combination therapy	I	Solid tumors and aggressive lymphomas	Completed	PR observed in 2 patients with ovarian cancer and 1 patient with DLBCL.	Fatigue (64%), headaches (50%), fever (45%), chills (45%), hyperbilirubinemia (34%)	Transient anemia (57%), hemagglutination (36%), lymphopenia (34%), thrombocytopenia (17%)
Sallman (Abstract; JCO, 2019) NCT03248479	5F9	5F9 in monotherapy or with azacitidine	I	R/R AML, untreated AML ineligible to induction chemotherapy, R/R MDS, and high-risk MDS	Recruiting	Objective response in 64% of patients with AML and 91% of patients with MDS.	Fatigue (21%) and infusion reactions (16%)	Anemia (38%), neutropenia (19%), thrombocytopenia (18%)
NCT04313881	5F9	Azacitidine (versus azacytidine alone)	III (double-blinded)	Untreated intermediate or high-risk MDS	Recruiting	N/A	N/A	N/A
NCT04435691	5F9	Azacitidine and venetoclax	Ib/II	Untreated AML ineligible to induction chemotherapy	Recruiting	N/A	N/A	N/A
NCT04778410	5F9	Cohort 1: azacytidine and venetoclax Cohort 2: mitoxatrone, etoposide and cytarabine Cohort 3: CC-486	I/II	Cohort 1: Untreated AML ineligible to induction chemotherapy Cohort 2: R/R AML Cohort 3: AML patients in CR or incomplete CR after induction chemotherapy	Recruiting	N/A	N/A	N/A
Ansell (CCR, 2021) NCT02663518	TTI-621	No combination therapy	Ib	Dose escalation: refractory lymphomas Dose-expansion: relapsed lymphomas, AML, MDS, and solid tumors	Completed	All patients with overt AML experienced progressive disease. One patient with AML and complete hematologic response developed complete remission. Another patient with AML and complete hematologic response maintained stable disease. Patients with MDS had stable or progressive diseases.	Infusion reactions (43%), chills (18%), fatigue (15%)	Thrombocytopenia (26%), anemia (13%)
NCT02641002	CC-90002	No combination therapy	I	R/R AML and high-risk MDS	Terminated after the dose-escalation portion	N/A	N/A	N/A

^#^ Objective response was defined as complete or partial responses. Abbreviations: DLBCL = diffuse large B cell lymphoma; AML = Acute myeloid leukemia; R/R = relapsed or refractory; MDS = myelodysplastic syndrome; CR = complete response; PR = partial response; NEJM = New England Journal of Medicine; JCO = Journal of Clinical oncology; CCR = Clinical Cancer Research; N/A = not applicable.

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
