# Peer review of "Harnessing Macrophages through the Blockage of CD47: Implications for Acute Myeloid Leukemia"

_cancers, 2021, doi:10.3390/cancers13246258_

Round 1
Reviewer 1 Report
In this paper, the Autors provide a comprehensive and up to date review of the therapeutical implications of CD47 blockade in AML and MDS.
CD47 is a well studied transmembrane protein which is normally expressed in healthy tissues and acts as a do not eat me signal by interacting with its ligand on phagocytic cells. Upregulation of CD47 has been described in cancer in general and in AML and MDS in particular and it is likely one of the mechanism exploited by leukemic cells for immune evasion.
Recently, some drugs targeting the CD47-SIRPa axis have been developed and have been tested in clinical trials in AML and MDS patients, with promising results.
The Author provide a clear and coincise overview of normal CD47 functions and its ligands, followed by a section highlighting the role of CD47 expression in AML.
In the last section of the review, they focalize on the trials evaluating CD47 blockade in AML.
Overall the review is very well written and clear.
Minor:
I suggest to add a figure describing the mechanism of action of the three drugs described (5F9, TTI-621 and CC-90002).
A table summurazing the ongoing and completed trials and their key results would also be beneficial in order to provide a rapid reference to the reader.
Author Response
Figure 3 has been added, showing the structure of the 3 drugs.
TAble 1 has been added, reviewing the clinical studies.
Reviewer 2 Report
This is a nice, timely and well written review.
I have a pair of suggestions:
a) Better discuss possible toxicities of CD47 targeting as the antigen is expressed by all normal cells.
b) Add a table summarizing characterisitics (for ongoing) and results (for published) of all clinical trials involving CD47 targeting, including Phase I development
Author Response
Side effects observed in the different trials have been included in table 1 which summarized the completed and on going clinical trials.